# Leaking Women: A Genealogy of Gendered and Racialized Flow

Michelle Fine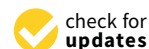

The Graduate Center, The City University of New York, New York, NY 10016, USA; mfine@gc.cuny.edu

**Abstract:** Through a feminist and critical race analytic, this paper theorizes the disruptions evoked by leaky women—actually doubly leaky women—those whose nipples, peri-menopausal uterus' and mouths have "leaked" in ways that rupture/stain/expose the white-patriarchal-capitalist enclosure of work, home and the streets and then dared to leak again by suing for justice in court. In a closing coda, I address the race/class policing dynamics between she who leaks and the "respectable" [usually white] women recruited to plaster up the hole and cauterize the leaker.

**Keywords:** women; silence; voice; leaking secrets; feminist genealogy

## 1. Introduction

In "Getting tense about genealogy," Daphne Meadmore, Caroline Hatcher and Ercia McWilliam (Meadmore et al. 2000) advance Foucault's call to contest "the legitimacy of the present . . . [which] can be undercut by the foreignness of the past, offering the present up for re-examination and further enquiry." (p. 464) The 'genealogical method allows the researcher to travel along rhizomatic pathways, searching for new vantage points from which to see the self. New vistas come into view . . . what is important is that the journey . . . rejuvenates and in doing so, offers new ways of seeing the present." (p. 474) Forty years earlier, Foucault argued " . . . the real political task in a society such as ours is to criticize the working of institutions which appear to be both neutral and independent; to criticize them in such a manner that the political violence which has . . . exercised itself obscurely through them will be unmasked, so that one can fight them." (Foucault 1974, p. 171).

So, what happens when women leak and go to court to insist on their right to leak in public? And how do understand how race and class, within gender, shape who leaks, who is punished for leaking and who is called upon to stuff the hole? Might we use such an analysis to "unmask" the political violence that disciplines such gendered *express*ions as disruptive?

In this essay I am interested in the ruptures, that is the leaks, that women release, carry, narrate, scream, drip onto the carpet of gendered and raced domination. Speaking both metaphorically and in ways embodied, about women's breast milk, blood, urine and wild tongues that violate norms, and drawing on an assemblage of archival material, I attempt to decipher how gender intersects with race and class in a watercolour pattern of who is accused of leaking, who is punished for leaking and who tries to discipline the leaky woman. She who is accused of leaking is disproportionately a woman of colour; she who is punished is typically a woman of colour and she who disciplines and polices her undisciplined sister is often white. The race:class theatre, within gender, is tragically but not entirely over-determined and deserves a critical pause and feminist reflection in deeply revolting times.

In conversation with Mary Douglas's (1966) stunning insights on purity and danger, as well as Seyla Benhabib (1993) and Nancy Fraser (1995) who theorize the fraying, ever porous and fragile membranes of public and private, I examine a series of gendered and racialized intrusions, by women and womanly fluids, dripping into public view. If read at an angle, these sister puddles coalesce along a "rhizomatic pathway," signalling how gender, always raced and classed, contains and conserves

political arrangements and how gendered leaks, by design, mistake or unapologetically, strip naked the normative violence and exclusions that curate women's lives (Collins 1998).

This argument is not crafted to argue that women leak more than men (although they probably do); that the gender binary is "real" (it is definitively not); that woman is a coherent category or that leaky women are punished more (although there is evidence that they are, see Miceli et al. 2008). In this article I seek to animate a dormant pattern, provoking what Maxine Greene (1977) would call a *wide-awakeness* to how women—mostly of colour—are charged with being out of order, mothering in ways inappropriate, labouring in ways undisciplined, speaking too bluntly, particularly when they pull back the curtains on the dirty understory of gendered, raced and classed "norms." As Meadmore and colleagues invite a rhizomatic pathway, in this essay I want to trouble the outrage evoked by women's leakage, instigating new ways of seeing the present, carving a critical space to think again about why/how/under what conditions women's leaks dis-ease. This pathway is of course carved idiosyncratically by me, others would navigate the material differently. A subjectively constructed pathway patched with feminist and critical race texts, the essay seeks to provoke a visible circuit of oppression and resistance, connecting breadcrumbs across social media and lawsuits, tracing race/gender/class where evidence allows.

Looking back in time and across a series of lawsuits, we will trace how women's bodies, fluids and words disrupt work spaces held together by gendered and racial hierarchies in drag as "professionalism," and we will explore how women have always resisted in secret language, in madness, in diaries, in writing, in protest and in court. Borrowing from the crafty language of Rebecca Solnit I am interested in the biography of women's "breaking stories," the price we pay for publicly and collectively interrupting stories that need to be broken. (Manne 2018; Solnit 2018)

*Pause for a Gendered Counter-Narrative*: As I choreograph the montage of "evidence" that (dis)organizes this essay we learn that public urinals are being installed on major walkways in Paris, to facilitate men traveling "to and from" work. While there is some local resistance, state policy has already designed and installed these "uritrollers" to facilitate the natural flow of male fluids, in public. State sponsored leaking for men.

According to Jessie Yeung of CNN, in 14 August 2018, "an attempt by officials in Paris to tackle public urination by installing open air urinals or uritrottoirs, has outraged some residents of the French capital. The new urinals, housed in flower boxes, aren't subtle- they're fully exposed on street corners, are painted bright red and have nearby signs advertising their presence." These are referenced as "intelligent urinals," and eco-friendly: equipped with a straw layer to eliminate odour and "harness nutrients in ways to produce compost for parks and gardens." The mayor of the fourth district of Paris tweeted they are "an invention of genius." We can only speculate about which men—elites? whites? French?—can enjoy the privilege of peeing in public during the day, while others risk criminalization (Yeung 2018).

But as we will see, across this funky rhizome, where there is oppression there is resistance. On 29 August, Angelique Chrisafis (2018) reports that protesters damaged 'sexist' open air urinals by "plastering them with stained sanitary towels and tampons, then blocking them with concrete . . . notes left behind attacked Paris authorities for encouraging men to unzip and relieve themselves without cover in open public spaces—even though public breastfeeding still elicits scorn."

Within spaces public, domestic and labour, women are typically expected to be containers for the run off of affect, pain, dependencies, male fluids, family and institutional excess. We are supposed to absorb secrets but not secrete. And yet we rise. (Angelou 1978)

I write in the era of Stephanie Clifford, aka Stormy Daniels, you may not remember her name. A white woman who admits an affair with #45 [Donald Trump], who received $130,000 in hush money, for a "catch and kill" story orchestrated by the National Inquirer, who went public with the arrangement, who is, with pride and accomplishment, an adult film star and producer; who has been degraded ruthlessly in the media and the courts.

I write in the era of the bold coming forth of Dr. Christine Blasey-Ford, white woman, neuroscientist, who testified to the sexual assault, 30 years ago, by now Chief Justice Brett Kavanaugh. Blasey-Ford and her family have endured death threats; they have had to move at least three times, with a lifetime trauma. Kavanaugh sits on the Supreme Court with a lifetime appointment.

I write in a time of Omarosca Manigault-Newman, a Black woman staffer with the Trump administration; a former Apprenticeship contestant, public relations/communication expert, a deep defender of the president against accusations of racism and misogyny. And then she was fired. She wrote a book called *Unhinged*—about the president—and her roll out was punctuated with audio and video "tell all." She is relentless—in an interview she told the newsman, "If you see me in a fight with a bear, pray for the bear."

I write in the mo(u)rning after the release of a major report indicting the Catholic Church in the U.S. for sexual abuse of children and institutional cover up and as activists report that the U.S. federal government has separated thousands of Central American children from their parents, at the border, seeking asylum. We don't know their names.

I write in the midst of #MeToo when men, wealthy and powerful, are finally facing the consequences of careers fortified by sexual violence, power grabs, assault, even as the president and his men continue to deny, grab and violate in public. Initiated by Tarana Burke, a Black woman dedicated to justice for all the women assaulted at work by powerful men, the movement quickly mobilized around the stories of famous white women actresses, almost eclipsing the stories of the most marginalized women of colour, poverty and immigration, followed, eventually by stories of men brutalized sexually at work, in the church, in Hollywood and by government officials. And by the time you read this, there will be more—and others in your own lands—whose names we know and don't; whose leaks were smothered, exposed, celebrated or humiliated.

Aware of course that women and men, cis, trans and genderfluid, are assaulted and silenced, with great disparities by race, ethnicity, class, (dis)ability and immigration status; aware of course that the systematic rape of enslaved and free women of colour, immigrant, indigenous, incarcerated or on the streets, is a long, state sanctioned violent stream of U.S. history not in the official record. Acknowledging that women in highly vulnerable positions—immigrants, domestics, nannies, cleaners, night time janitors, sex workers—are routinely threatened and assaulted and we hear not a peep. I write to honour all and to make visible those who leak and to document what happens after the flood. I am interested in how and when women "leak" from our many dangerous orifices—vagina, nipples and mouths—and how other women are called in to stanch the flow.

## 2. Results

### 2.1. Let it Bleed

You may remember . . .

"It's disgusting," remarked Donald Trump, then presidential candidate, attacking FOX TV host Megyn Kelly, "You could see there was blood coming of out of her eyes, blood coming out of her wherever."

Women's blood has long terrified men.

Joan Chrisler (2011) pens a brilliant excavation of the psychological literatures on the threat of the menstruating woman. Jane Ussher (2006) theorizes "the power and danger perceived to be inherent in woman's fecund flesh, her seeping, leaking, bleeding womb standing as site of pollution and source of dread," (1). Richard Stockton in the "people's history of menstruation" (Stockton 2018) reminds us that fear, shame and restraint of menses has a long history. In Ancient Egypt menstrual blood was gathered for spell casting and medical treatments; the Greeks blended menses with wine to enhance the harvest. Turning toward religion, we learn that the Talmud addresses "niddah" as the period of woman's uncleanliness, requiring physical separation and ritual cleansing before a woman can return

to touch people and things. Islam restricts the prayer traditions women can practice as they menstruate and apparently "men are not permitted to divorce their wives during that time of the month."

Lucia Bushak, in "A brief history of the menstrual period: How women dealt with their cycles throughout the ages," (Bushak 2018) reminds us that in ancient and medieval times, female "bleeding without injury" was traced to the moon, considered "holy and mystical, at other times cursed and untouchable." Bushak introduces Pliny the Elder, a Roman author and philosopher, who believed nude menstruating women could prevent hailstorms and lightning, while frightening insects away from crops. Mayas believed menstruation to be a punishment from the Moon Goddess after she slept with the Sun god. Sara Read tells us that women used rags or other fabric (hence, on the rag) and soiled their clothes. In medieval Christianity, women adorned herbs to overwhelm the smell of blood but "pain relief was not readily permitted by the Church: God apparently wanted each cramp to be a reminder of Eve's Original Sin."

By 1879, a crude tampon like insertion device was developed, as was the Hoosier sanitary belt. By the late 1880s, pads began to appear. The first official tampon was developed in 1929. Women were asked to place money into a box instead of handing money to a salesperson. As women's "natural" flows moved into the public sphere, stigma and shame amplified. (see Chrisler 2011, for a rich comprehensive review of literatures on menstruation, shame and stigma).

To date, across the globe, a large swath of girls and women are denied access to simple sanitary products, an extension of the racialized, classed and caste-based layers of reproductive, economic and educational (in)justice. In India many girls and women lack basic sanitation and sanitary products, limiting access to school and work. As Inga Winkler and Virginia Roaf (Winkler and Roaf 2015), in "Taking the bloody linen out of the closet," frame menstrual justice as a human rights issue, recognizing: "The taboo and silence around menstruation makes menstruation a non-issue . . . [and yet] the silence and shame are themselves violations of human rights and threats to public health." They report that Rajasthan and Uttar Pradesh, India, in which more than half of girls had no information about menstruation prior to menarche.

Menstrual silence and stigma feed repulsion, detected even in the psychology laboratory. In a social psychological experiment conducted in the U.S., Toni-Ann Roberts, Jamie Goldenberg, Cathleen Power and Tom Pyszczynski (Roberts et al. 2016) found that after dropping a tampon (as opposed to a barrette) visibly on the floor, women were rated as less competent, less liked and subjects had a marginal tendency to physically distance from her. By integrating a range of evidence and situating menstrual hygiene within a human rights framework, Winkler and Roaf elaborate the conditions and consequences of menstrual silence, shame and ignorance as they undermine both human dignity and the right to privacy, limiting education, travel and employment, with particularly severe consequences for homeless women, women living in informal shelters, women in prison, women with disabilities and/or sex workers.

Along the rhizomatic pathway, one dynamic is predictable: where there is silence and repulsion, there is resistance. On 30 May 2018, Scotland announced an initiative to provide free sanitary products to low income girls and women:

> "Women from low-income homes across Scotland will be offered free sanitary products as part of a pilot scheme in one city is being rolled out across the country. The trial project in Aberdeen last year was funded by the Scottish government and distributed free products to more than 1000 women. Now ministers will provide charity FareShare with more than £500,000 to extend the scheme which aims to reach more than 18,000 more people." (Yeginsu 2018)

Back in the United States, after Alisa Coleman was fired for staining a chair at her place of employment with unexpected peri-menopausal blood, she sued in Alisa Coleman v. Bobby Dodd Institute (2017), heard in the United States Court of Appeals for the Eleventh Circuit:

Plaintiff-Appellant Alisa Coleman, an E-911 Call Taker at Bobby Dodd Institute, a job training and employment agency located in Fort Benning Georgia that serves people with disabilities. In August 2015, Ms. Coleman "unexpectedly experienced a sudden onset of her menstrual period that resulted in her accidentally leaking menstrual fluid on her office chair. She reported the event to her supervisor who told her to leave work and change clothing. Two days later she received a disciplinary write up "that she would be fired if she ever soiled another chair from sudden onset menstrual flow." An African American mother, who had worked at the call centre for close to a decade, took measures to assure this would not happen again but in April 2016 she rose to walk to the bathroom and some fluid stained the carpet. "She immediately cleaned the spot with bleach and disinfectant." On 26 April she was terminated for her alleged failure to "practice high standards of personal hygiene and maintain a clean, neat appearance while on duty." She "soiled herself and company property." Drawing on Title VII of the Civil Rights Act and the Pregnancy Discrimination Act, Coleman's lawyers sued for sex discrimination, arguing that, "Just as the PDA prohibits discrimination on the basis of a woman's capacity to become pregnant, see Johnson Controls and the perceived normalcy of a woman's menstrual cycles, see Harper, the PDA must also be understood as encompassing the cessation of a woman's reproductive capacity through the physiological process of peri-menopause and ultimately menopause." By November 2017, Bobby Dodd Institute settled with Ms. Coleman.

And in the spirit of punishing women for carpet stains, recall the brave Nafti Diallo: Excerpts from an article in NEWSWEEK magazine are captured in italics (Dickey 2011):

*"Hello? Housekeeping." Diallo glanced around the centre room of the hotel suite. [a} naked man with white hair caught her by surprise. "Oh my God, I'm so sorry." "You don't have to be sorry . . . " He grabbed her breasts. He slammed the door . . . "Sir, stop this, I don't want to lose my job," . . . .*

*"You're not going to lose your job."*

*"He pulls me hard to the bed . . . He tried to put his penis in her mouth . . . Diallo kept pushing him away, "I don't want to hurt him . . . I don't want to lose my job." He shoved back, moving her down the hallway from the bedroom toward the bathroom . . . Strauss-Kahn . . . pulled up [her uniform] around her thighs and tore down her pantyhose, gripping her crotch so hard that it was still red at the hospital, hours later. He pushed her to her knees, her back to the wall. He forced his penis into her mouth, she said and he gripped her head on both sides. "He held my head so hard here, . . . he was moving and making a noise. He was going like 'uhh, uhh, uhh.' He said, "suck my—I don't want to say." The report from the hospital where Diallo was taken later for examination notes that 'she felt something wet and sour come into her mouth and she spit it out on the carpet."*

*"I got up" Diallo told NEWSWEEK. "I was spitting. I run. I run out of there. I don't turn back. I run to the hallway. I was so nervous; I was so scared. I didn't want to lose my job."*

*[S]he hid around the corner in the hallway near the service lobby and tried to compose herself. "I was standing there spitting. I was so alone. I was so scared." . . . .*

*NEWSWEEK continues, "Many aspects of Diallo's account of the alleged attack are mirrored in the hospital records, in which doctors observed five hours afterward that there was 'redness' in the area of the vagina. and pain to the left shoulder. Weeks later, doctors re-examined the shoulder and found a partial ligament tear . . . DNA evidence in suite 2806—the result of all that spitting that mingled the maid's saliva and Strauss-Kahn's sperm—makes it virtually impossible to deny there was a sexual encounter between Dominique Strauss-Kahn and Diallo."*

\*\*\*

Dominique Strauss-Kahn was the managing director of the International Monetary Fund. At age 62, he was also the much-touted potential presidential candidate in France. Nafissatou Diallo was a maid in the Hotel Sofitel, mother of a 15-year-old daughter, age 32, a Guinean asylum seeker living in the Bronx.

The media chronicled her "lies"—many of which are simply evidence of what it means to live in the gendered, classed and racialized world of immigration, poverty and insecurity in New York City. She is accused of lying about a gang rape in her home country which enabled her to establish asylum-worthiness in the U.S. Apparently, she "hangs out" with "unsavoury characters" including men who are in prison for dealing drugs. She may have falsified some tax documents, so she would be entitled to additional welfare payments. But no one disputed what happened in the room.

At a press conference called by Diallo's supporters in July, demanding that the District Attorney take the case, the room filled with African American and African men and women stood in solidarity with Diallo's right to go to trial. Strauss-Kahn's lawyer asked if the carpet were tested for semen. An obviously frustrated man who introduced himself as a retired police officer and member of One Hundred Black Men in Uniform, responded: "All we are saying is to take this case to trial . . . Your question, however, I find offensive. Would you have asked those questions of the Central Park jogger case, where the woman was White and the accused boys were Black? Asking for the medical records is like demanding to see Obama's birth certificate. . . . these are the materials that would be submitted during the trial."

On 23 August 2011, the New York City District Attorney Cyrus Vance decided to drop all attempted rape charges against Strauss-Kahn because Diallo was considered "not credible." They engaged in a "quick consensual sexual act" that morning. He never spoke; he never had to. But his wife did: Anne Sinclair, Strauss-Kahn's wife, an elite white woman pitched against a Black immigrant hotel worker, allegedly told a friend that her husband "is a seducer, not a rapist" and that he "had" three women that weekend, in preparation for his presidential bid.

In July 2011, Tristane Banon, a young white journalist from France filed a formal complaint against Strauss-Kahn in France for an alleged attack in 2002. Banon originally dropped the charges because her mother, who worked with Strauss-Kahn and admits that she also had "consensual but rough sex" with him. After a civil suit by Banon and Diallo, on December 2012, the former IMF chief settled with Diallo for an alleged $6 million dollars. Diallo's credibility was enhanced only when confirmed by a young elite white woman, Banon.

As Dorothy Roberts (1999) argues in "Killing the Black body" and Kate Manne (2018) in "Down girl, the logic of misogyny," the logic and abuse embedded in misogyny, heterosexism, class and racial hierarchy have been normalized in state, corporate and domestic institutions. Racism accelerates the fierce and vengeful temper of misogyny, whether it be in the case of Alica Coleman or Nafissatou Diallo.

When we secrete—through nipples, vagina and mouth—women need to be punished, stuffed and silenced, because we/they reveal what Collins (1998) would call the matrix of domination that coheres domestic, political and work life. Race and class dynamics differentiate *how* women are supposed to sustain organizational life, *how and if* they leak and how their leakage will be managed—by men and by other women. Women of colour are often exposed to the raw dirty underside of domestic/organizational life and in the process are exploited, ask to "hold"/whet nurse the secrets of white supremacy, labour, gender, carceral state, gentrification, picking up the pieces of capitalism/racism/patriarchy as Powell Pruitt (2004) would argue and then punished when they speak or leave evidence on the chair or the carpet.

*2.2. Leaky Nipples*

And then there are the waves of lawsuits surrounding nipples: women breastfeeding their babies in public. In 2017, The Centre for Work Life Law at University of California Hastings College of Law reported an 800% increase in breastfeeding related discrimination lawsuits over the past decade, from "nursing mothers and babies being kicked off planes to a mom who attempt to protect her breastmilk

from TSA screening." (Centre for Work Life Law 2017) To the extent I can determine, these appear to be largely cases brought by white women with flexible/secure enough jobs to enable a micro-second of privacy and autonomy, combined with a rich sense of entitlement and then denied.

Consider: In 2012 Stephanie Hicks, a white woman police offer with the Tuscaloosa Police Department, was told to pump in the locker room. Fellow officers made radio calls to "wrap those boobs up" accompanied by write ups, demotion, reduced pay, late shifts and poor performance evaluations. Lieutenant Teena Richardson, Hicks' immediate supervisor, admitted she was bothered by Hicks' twelve weeks of Family Medical Leave Act; Richardson had recommended six. Richardson was also offended that Hicks was given a desk job to interrogate pharmaceutical fraud cases to avoid "on call" duty.

Prior to her leave, Hicks received an evaluation that she "exceeded expectations" but was written up first day back. Hicks overhead Richardson referring to her as "that b****" and complaining that "stupid c*** thinks she gets 12 weeks but I know she only gets six."

Within two months of returning to work, Hicks sued for hostile work environment and was awarded $374,000 in damages. Drawing on the Pregnancy Discrimination Act (PDA) the jury "decided that Stephanie Hicks suffered discrimination in violation of the PDA and retaliation, in violation of FMLA, when she was reassigned only eight days after returning . . . following childbirth. The City's failure to accommodate Hick's requests, when it allowed accommodations to others similarly situated, constituted discriminatory constructive discharge, in violation of the PDA. We find that a plain reading of the PDA covers discrimination against breastfeeding mothers. This holding is consistent with the purpose of PDA and will help guarantee women the right to be free from discrimination in the workplace based on gender specific physiological occurrences."

Forty-seven states, Washington DC and the U.S. Virgin Islands have laws that specifically allow mothers to breastfeed in any public or private location. South Dakota and Virginia will not prosecute breastfeeding moms for public indecency or nudity laws. Idaho is the state that has yet to pass any laws to protect breastfeeding mothers. In fact, the only legal protection in Idaho is jury duty exemption. The Idaho Breastfeeding Law Coalition initiated a petition, gathered 1400 signatures but Idaho is not close to passing legislation.

For a website of breast-feeding shaming incidents see: https://www.huffingtonpost.com/2014/08/01/map-where-its-legal-to-breastfeed-in-public_n_5637301.html?slideshow=true#gallery/210867/0.

Breastfeeding at work is a vibrant movement, mobilized largely by elite women—and yet the surveillance and invitations to move out of public view, persist. On 24 January 2019, *People magazine* featured of picture of a white mother double breast feeding her 18-month-old twins, breasts full and descended and with a defiant fist in the air, under a headline, "Breastfeeding mother of twins is told to use a private room in her children's day care." (Juneau 2019). Recall the story told by Lorde (1980) about the oncology nurse who insisted that Lorde insert a prosthetic breast, least she upset the other patients.

We are reminded again and again by these stories of women's body parts and fluids shamed—deemed out of place, sent away or normalized so as not to disrupt. But the question lingers—who/what is being disrupted? To help us understand, we resurrect Mary Douglas' brilliant analysis, "*Where there is dirt there is system. Dirt is the by-product of a systematic ordering and classification of matter.*" (p. 37)

*2.3. Wild Tongues*

"wild tongues can't be tamed, they can only be cut out" Anzaldúa (1987)

It is not only fluids, of course, that leak but mouths—another dangerous orifice if attached to a woman's body. In the 2018 edition of the *Columbia Journal of Gender and Law*, Clare Tilton examines gender and whistleblowing, particularly the costs, financial risks and vulnerability employees incur after reporting ethics violations within their organizations. Tilton (2018) defines whistleblowers as

"organization members . . . who disclose illegal, immoral or illegitimate practices . . . to [those] who may be able to effect action." Thinking through gender politics, she argues that "employees who fear retaliation from superiors may turn to external reporting in an effort to find protections they do not see within their organizations." (vol. 35, p. 342) While evidence is contradictory on the frequency, motives and strategies for how women (white and women of colour) blow whistles, there is substantial anecdotal and increasingly empirical evidence that women experience themselves, more so than men, as organizational outsiders, occupy positions of low power and run the social risk of retaliation for betrayal and for acting against gender norms. And there is cumulative evidence that whistle blowing by leaders tends to be celebrated, while whistle blowing by subordinates (the roles women/people of colour/queers and all who overlap, are more likely to play) tends to be vilified.

Marcia Miceli et al. (2008) co-authored a national study of women and men on an Air Force base, to assess who and why people blow whistles. Of the 3288 employees who responded, 37% reported they observed wrongdoing and 26% reported the wrongdoing. Of those who reported wrongdoing, women experienced more retaliation for reporting than men, regardless of rank. In fact, neither women's status nor range of authority protected them from retaliation. And yet, even after retaliation, women were more likely than men to report the original wrongdoing to an outside agency. Once and then twice wronged, they were not deterred. The authors conclude, "The more retaliation they faced, the more likely women were to keep fighting the battle over what they felt was wrong."

Tanya Kateri Hernandez (2000), in her essay Sexual Harassment and Racial Disparity, published in The *Fordham Law Archive of Scholarship and History*, digs into the questions of whistle blowing, harassment and retaliation for sexual harassment claims advanced at the gender/race intersection. Drawing on EEOC statistics on sexual harassment charges in the United States, Hernandez finds a disproportionate overrepresentation of women of colour and conversely a disproportionate under-representation of White women in the charging parties. The race-based disparity has been interpreted in two ways: women of colour disproportionately experience harassment and/or women of colour disproportionately report it when it occurs. Hernandez argues persuasively for the former position: "sexual harassers target White women as victims at disproportionately lower rates than women of colour." (p. 187) While various factors influence vulnerability, coping strategies and likelihood of reporting, the bulk of the evidence points to the conclusion that women of colour endure more harassment than white women. While the question of who endures more is both unanswerable and absurd, it is absolutely clear that women of colour and queer women, more boldly, more publicly and more collectively resist as, to borrow from Sara Ahmed, "wilful subjects." (Ahmed 2014).

And yet when women go public with critique, men—but also women—can be found rushing to defend the disciplining structures under question. Recall the backlash experienced by Alice Walker, after she published *The Color Purple* (Walker 2007). The book opens: "You better not never tell nobody but God. It'd kill your mammy." Celie, age 14, raped by her stepfather, was writing a letter to God. In an interview with *The Guardian*, Walker speaks of years of reputational abuse she endured, in response to the violence portrayed in the novel. She explains that even and especially from within her own community, there was a desire to smother evidence of gendered violence within the Black community. Walker wrote of "batterers, womanisers, alcoholics, all of that." Surprised by the aggressive demand to silence her, she reflected, "You'll notice that most people, in discussing [Mister] who was son of a slave owner, they just cut it off right there, they act like 400 years of being dominated and enslaved by white men left no trace and that all this bad behaviour started with the black people. It's so ridiculous. But it's the way that people distance themselves from their own history and their own participation in what is very bloody and depressing behaviour, over centuries."

According to the interview, Walker was accused by women and men of "betraying her race, of hating black men, of damaging black male and female relationships . . . " And yet she explains that she wrote the novel "to support men and women who are in abusive relationships you know? Who are trying to figure out how we got into this position where, after, you know, 400 years of slavery we're still treating each other like slaves." (Walker 2003).

## 2.4. Resist(er)ing

Wandering this rhizomatic pathway, we bear witness to sustained historic and current assault on individual women who dare and the relentless resistance by collectives. Even in the most horrific circumstances, even and particularly from women who seem to have the most to lose, women (again, cis, trans, genderfluid) engage fierce resist(er)ance. Fifteenth century Japanese women crafted a secret language; "witches" have long concocted spells in the quiet gendered spaces for revenge. Consider the practice of Nushu, the 19th century Chinese script only women could write, practiced by peasant women to communicate with "sworn sisters" in Shanjiangxu (Young 2017) or women's diaries, particularly Black women's diaries as Mary Helen Washington (Wells-Barnett 1995) wrote in the foreword to *The Memphis Diary of Ida B. Wells*, "Every woman who has ever kept a diary knows that women write in diaries because things are not going right." Washington continues, "I think of the diary as something like the Clearing in Black religious culture, a place where, physically and psychologically, Black people felt free to speak in a setting outside the boundaries of the official church, a private sanctuary where one's truer self is affirmed and authorized."

## 2.5. Coda: When Women Police Each Other

For all the presumed sisterhood in resistance, I offer a coda on women policing each other—across the stratified fences of race, class, sexuality, disability, documentation status—but mostly, here, race. There is too much history, ample empirical matter and anecdotes that coagulate into shared observation: that women—often elite, white, heterosexual and married—are recruited and/or volunteer to attack and invalidate the leaky woman, especially women of colour or white women who threaten hetero-patriarchal hierarchy as manipulative, lying, gold diggers or "not credible."

This is, of course, an ugly (well known) counter-narrative. Sarah Brazaitis (2004) helps us understand this inconvenient but highly predictable truth. Brazaitis writes on white women's particular valence to position them/ourselves in groups, social relations and organizations, to protect and maintain the status quo of race and hetero-gender relations. Aida Hurtado has written that White women are seduced into being the partners of white men under the pretence of sharing power and yet "The patriarchal invitation to power is only a pretend choice for white women . . . because their inclusion is dependent on complete and constant submission" (Hurtado 1989, p. 845). Bell and Nkomo (2001) interviewed 825 Black and white women managers about gender and race dynamics at work and found "White women are quick to come to the defence of their white male superiors often acting as their talking heads, echoing and supporting their views and values to fellow workers." Put more pointedly, Connolly and Noumair (1997) have written, "White women have been used as a prophylactic against interrupting patriarchy." (p. 331) Brazaitis, a group relations theorist, extends the argument: when "white women are passive and docile, Black women are left with rage and aggression." (p. 110) Another fragment of evidence: analysing the 2012 American national election study, Christopher Sout, Kelsy Kretschmer and Leah Ruppanner, (Sout et al. 2017) found that married white women and Latinas have significantly lower levels of agreement that "my fate is linked with other women" than unmarried women of the same ethnicity. Interestingly, they find no such relationship for Black women.

Back to the 53% of women who voted for Trump, a brazen and proud racist, misogynist, Islamophobe, homophobe, xenophobe and policy sadist. I wondered, then and now, who they are? I have learned a bit. White working class without college—well yes, 61% of White working-class women without college voted for Trump but so did 44% of White women with college, after seeing the tape where he proudly talks about being able to "grab women by the pussy." Education matters for white women. In contrast: college made NO difference for Black and Latino women. Across educational levels, a full 94% of Black and 68% of Latin women voted for Clinton. Same for lesbians, bisexuals and transwomen. The LGBTQIA community voted overwhelmingly for Clinton (again, I am not romancing Clinton, but they certainly didn't vote for Trump). Thirty two percent of unmarried women (sexuality unspecified) and 49% of married women voted for Trump and 53% of white women (see important critical heterosexuality studies schools, Wittig, Butler, Mary Holmes, Chris Beasley,

Heather Brook for more . . . ) Age matters surely. But I want to lift up a discomforting question, culled from these fragments of evidence, about women snuggling under the covers of Whiteness, economic comfort and heterosexuality—marriage in particular—at work and at home. Under what conditions do white heterosexual married women leave their feminist consciousness at the bedroom/corporate door? (I hear the hauntings of Adrienne Rich "I told you this in 1980!").

Let me leave you with one other claim: She/they who transgress "woman" will be punished; but she/he/they who resist and disrupt gender/sexuality norms, particularly young people of colour, are under chronic scrutiny and corporeally vulnerable to the very people who are supposed to "protect" (see Nielsen, Walden and Kunkel, (Nielsen et al. 2000) 15 year analysis of gendered heteronormativity and social reactions to gender norm violations). In our national, participatory study of more than 6000 queer youth (disproportionately of colour, out of school, trans, system involved), *What's Your Issue*?, we have gathered substantial evidence that queer youth, trans youth and those who refuse gender binaries, particularly of colour, are more severely policed and punished by public authorities—in schools, by police, in public housing, health care and on the streets—than cis and/or white youth. Consider the words of Elana, a Puerto Rican lesbian 11th grader who explained to a large focus group of LGBTQIA/GNC youth discussing New York City policing: "When I walk down the street alone I am fine but when I walk arm in arm with my girlfriend police say to us, "I want to fuck both of you." The group snapped their fingers in solidarity—and painful recognition (Fine, Torre, Cabana, Frost, Avory, (Fine et al. 2018)). Being policed and violated by the state is not a metaphor but an existential threat, for those who transgress gender/sexuality categories, particularly transwomen of colour.

### 2.6. On Flows, Floods and Fissures

Leaky women have been burned at the stake; institutionalized; beaten and killed by men and by their mothers; humiliated and attacked by other women; discredited in the media and sexualized/violated by police (see Manne 2018). But the rage, critique and the stubborn desire will not be extinguished. Some went mad and scratched the *Yellow Wallpaper* in the attic (Gilman 1892); some ran away, only to find the violent arms of another man; some sued and lost; a few sued and won. Today we/they leak and rise in staccato rhythms, in puddles across the globe—in workers' rights movements, domestic violence struggles, reproductive justice mobilizations, in the Fight for 15, for socialism, for literacy, for queer, immigrant, Black Lives Matter, environmental and indigenous justice. No longer (only) swallowing or whispering their concerns; no longer simply and slowly going mad; no longer only running into the arms of another exploitative man. Leaking women are disrupting the category woman, insisting on economic, gender, racial, immigrant, environmental and reproductive justice; seeking not assimilation but radical transformation. Young deliciously leaky and unapologetic activists commit to wild solidarities, linking across struggles for queer, Muslim, undocumented, Black Lives Matter, indigenous, labour, environmental and economic justice and prison abolition over time, space and social media. The leaks become floods. Even as public urinals are "erected" in Paris, the halls of the U.S. Congress fill with trans women, veterans, women in hijab, breast feeding, women of all colours, lesbian and bisexual, 90 year olds and 29-year olds. But before we celebrate women (trans and cis) publicly leaking, breaking silence and linking arms, we must interrogate the relentless race/class dynamics within this ever contested category woman, address racial policing within and among us, theorize the response-ability of intersectional, transnational feminism as a liberatory and not another colonial, project.

**Funding:** This research received no external funding.

**Conflicts of Interest:** The author declares no conflict of interest.

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
