# Peer review of "Leaking Women: A Genealogy of Gendered and Racialized Flow"

_genealogy, doi:10.3390/genealogy3010009_

Round 1
Reviewer 1 Report
I love the premise, but the abstract is a little confusing. Can you be more clear about what you are arguing there?
Also in the introductory paragraphs unpack what it means to leak and also why certain women are seen as leaking more--this paper doesn't seem to acknowledge as much as it should differences in class, race, ethnicity, age or gender in the introduction, although it is clear later on that this is what you are doing. It seems as though you get to the meat of the article with the discussion of the public urinals. You might add that certain men have always felt privileged to urinate in public with no repercussions from early childhood on.
The discussion of your mother seems kind of tangential, especially since she was busy policing herself not to leak in any way. Or maybe you can align your mother more clearly with the women that do police other women.
line 258 Diallo WAS considered not creditable. Add was.
This is a lively and interesting essay that I think the readers will enjoy if they can get past the first two pages, which don't really tell us where the essay is going. A better abstract and a more clear introductory paragraph would take care of that.
Author Response
Thank your very much for your review.
The text has been revised, taking into account:
1. I have removed "horizontal genealogy" and replaced with rhizomatic pathways which was more fitting and less confusing
2. as suggested I have more directly relied upon feminist theorizing and critical race theorists including Fraser, Douglas, Ahmed, Hill Collins
3. I have removed the autoethnographic material in the introduction and shaved back much on the "voting for Trump" section
4. I have cleaned up the citations within the text, and in the bibliography
5. I have most dramatically reframed the introduction for clarity and to provide a road map
6. I have more explicitly created an intersectional analysis - working through gender, race and class as the archival materials permitted
I hope you find the current text satisfactory.
Reviewer 2 Report
The author starts with several quotes from Foucault which indicate her approach and then contradicts her own argument by stating she wants to offer a "horizontal genealogy".
This is a contradiction in terms and not supported by Foucault's approach. Genealogy is a diachronic method looking beyond cause-effect and at histories in the plural in terms of discourses, practices and institutions. It cannot be horizontal in the sense of examining synchronic phenomena which is what the essay does. Nor can the actual discussion of horizontal/synchronic events in recent years work as she is clearly using examples where hierarchies are key to all explanations offered and these are never horizontal situations because "power" and "resistance to power" are always happening in hierarchical structures of institutional power as is demonstrated.
The personal story from the author's own life offered at the beginning as justification for the approach does not contribute to the overall argument because it does not explore the context for silencing and telling sufficiently well throughout in a consistent fashion. Alice Walker's quote from one interview about responses to the Color Purple match this part of the narrative but the rest of the argument does not. There is an interesting argument to be made about confession/silencing, telling your friends/family as opposed to speaking publically but this is not pursued as the purpose of the essay. This does relate to Foucault's arguments about the visibility of the discourse on sex/ sexuality. It does not relate to Foucauldian feminist scholars on the law like Nancy Fraser or Seyla Benhabib - both of which could have been explored or whose approaches might have been cited or discussed.
Part of the problems with this text are the inclusion of so many diverse examples: men's urinal leakage in Paris, a man's rape/sperm as leakage on a carpet (rather than Diallo's spit/disgust) and women's resistance or discredited statements to this are not sufficiently contrasted to the examples of women's actual leakage of bodily fluids in menstruation and breastfeeding court cases in the USA. It would require a book length work to do sufficient justice to the range offered here. How do these particular examples demonstrate that an intersectional approach to
justice can be adopted in US courts or indicate the rise or fall of
instances of bias in court judgements or employers/employee tribunals? As such, the essay reads like early 1970s radical feminist arguments, again, Mary Daly or Shulamith Firestone, which were criticised for the range and imprecision of their examples of a totalising unity about women. This was tied to the failure of arguments about sisterhood to encompass all women.
It would have been better to follow in depth one subject and discuss the decision making processes of the court on different legal cases on this one subject and focus exclusively on the USA as the case law explored. The section on voting as leaking(?) does not work in the context of the argument and, if it is the case that married women do not reach out or identify with their sex/as other women, what is the argument offering that revises feminist work on this? Patricia Hill Collins is elsewhere endorsed but not cited in the references.
Most of the examples come from 2000 or later indicating no genealogy
is offered with any depth. She does not examine how social stigma on
visible menstruation blood at work is turned into a breach of code of
practice/conduct at work - which is an interesting argument - but surely there are more precedents for this than one case. To explore a range of these on this topic would have made this essay more promising, especially if she could have tied in court cases from different ethnic/religious US communities and different periods of recent US history over the post-war period even to add weight to her very strong multicultural approach. The examples offered are not new empirically and only offer recent media discussion as evidence or discuss other people's surveys. This could work as an approach if the examples had depth of analysis but not in this case.
The only attempted genealogy is of menstruation and the source here is very shaky - especially given the extensive feminist literature on women's menstruation, which it would have been interesting to site, especially on pollution and shame, e.g. Germaine Greer on the menopause or Mary Douglas on pollution. It reads like a wikipedia page entry which is insufficient for a scholarly text.
In conclusion, the essay needs refocusing and rewriting with careful consideration of what a genealogical approach can and can't offer the study of case law of the leakage of female bodily fluids in public spaces.
Author Response
Thank your very much for your review.
The text has been significantly revised, taking into account:
1. I have removed "horizontal genealogy" and replaced with rhizomatic pathways which was more fitting and less confusing
2. as suggested I have more directly relied upon feminist theorizing and critical race theorists including Fraser, Douglas, Ahmed, Hill Collins
3. I have removed the autoethnographic material in the introduction and shaved back much on the "voting for Trump" section
4. I have cleaned up the citations within the text, and in the bibliography
5. I have most dramatically reframed the introduction for clarity and to provide a road map
6. I have more explicitly created an intersectional analysis - working through gender, race and class as the archival materials permitted
I hope you find the current text satisfactory.
Reviewer 3 Report
This paper makes a good argument for inclusion as a genealogy of women’s leakage and the subsequent dangers thereof. The paper is well-structured, although there are disruptions in the variations of paragraph lengths. Given the content of the paper and the argument of the dangers, (im)possibilities and problematics of speaking/leaking – this disruption does not feel inappropriate. The author’s voice leaks onto the page with an essence of humour in places, contrasting with the horrors of women’s experiences and the consequences of leaking. I especially love the conclusion.
The article requires a thorough edit however and there are areas where further work on referencing should be considered before final submission, especially in regard to the consistency of the inclusion of the year of publication of cited authors in places. I have listed some of the punctuation, grammar and possible typos that require checking below.
73 – “in was to” - in ways to? (check quote)
258 – was considered?
275 –276 Capital K for kate
301 – Richardson recommended?
p. 307. Extra full stop after ‘childbirth’
314 - … after clear and does forty require a capital F?
402 – unintended? Just checking
424 – grammar – have
430 – “not going rights” Check wording of the quote
504 – 508– check for grammar and typo here. I’m wondering if breaking down this sentence would help with accessibility.
Author Response

(The authors gave the same response as above.)

Round 2
Reviewer 2 Report
I appreciate the effort that has gone into rewriting this piece.
It has coherence as an article now. The addition of greater multi-cultural examples and references helps but it is not a fully inter-sectional approach.
The metaphor of "rhizomatic pathway" should be unpacked or modified a little further. It is your navigation - ie taking a path -through an existing rhizomatic structure - certain social codes - that this essay explores.
Other people using the same examples would draw a different path or link them together using other paths of enquiry. The rhizomatic structure is your construction upon the range of selected diffuse and disparate examples of bodily fluids becoming subject to legal casework or public protest or stories in the news.
Author Response
i have clarified that this rhizomatic pathway is crafted subjectively and that others would / not connect the dots as i have; and have tempered the intersectional claims, recognizing that many of the 'cases' are without information about race and class, and so those connections are also provisionally woven into the text. on page 2 i have added
This pathway is of course carved idiosyncratically by me, others would navigate the material differently. A subjectively constructed pathway patched with feminist and critical race texts, the essay seeks to provoke a visible circuit of oppression and resistance, connecting breadcrumbs across social media and lawsuits, tracing race/gender/class where evidence allows.